# Comprehensive Phytochemical Profiling of *Ulva lactuca* from the Adriatic Sea

**DOI:** 10.3390/ijms252111711

**Published:** 2024-10-31

**Authors:** Zorana Mutavski, Igor Jerković, Nada Ćujić Nikolić, Sanja Radman, Ivana Flanjak, Krunoslav Aladić, Drago Šubarić, Jelena Vulić, Stela Jokić

**Affiliations:** 1Institute for Medicinal Plants Research “Dr. Josif Pančić”, Tadeuša Košćuška 1, 11000 Belgrade, Serbia; zmutavski@mocbilja.rs (Z.M.); ncujic@mocbilja.rs (N.Ć.N.); 2Faculty of Chemistry and Technology, University of Split, Ruđera Boškovića 35, 21000 Split, Croatia; igor@ktf-split.hr; 3Faculty of Food Technology Osijek, Josip Juraj Strossmayer University of Osijek, Franje Kuhača 18, 31000 Osijek, Croatia; ivana.flanjak@ptfos.hr (I.F.); kaladic@ptfos.hr (K.A.); dsubaric@ptfos.hr (D.Š.); 4Faculty of Technology Novi Sad, University of Novi Sad, Boulevard cara Lazara 1, 11000 Novi Sad, Serbia; jvulic@uns.ac.rs

**Keywords:** green macroalga, volatile organic compounds, GC-MS, fatty acids, amino acids, pigments, Q-TOF

## Abstract

The potential of the green macroalga *Ulva lactuca* is increasingly recognized, not only for its environmental benefits, but also for its applications in various industries, including food, pharmaceuticals, and cosmetics. Given this insight, a comprehensive analysis of the chemical profile of *U. lactuca* from the Adriatic Sea was carried out. The hydrodistillate, rich in (*Z*,*Z*,*Z*)-hexadeca-7,10,13-trienal and hexadecanoic acid, underlines its importance for health-related uses, particularly in lipid metabolism and cellular integrity. Fatty acid analysis showed a predominance of palmitic acid and a favorable n-6/n-3 polyunsaturated fatty acid ratio, suggesting that *U. lactuca* can make a valuable contribution to a balanced diet. In addition, essential amino acids, including leucine, valine, and isoleucine, support its use as a functional ingredient for muscle repair and metabolic health. The ethanol extract contained 56 compounds, including derivatives of fatty acids, phenolic acids, pigments, flavonoids, and steroids. Many of them, such as hexadecasphinganine, azelaic acid, 5-sulfosalicylic acid, and pheophytin *a*, have proven roles or potentials in promoting human health. These results confirm that *U. lactuca* is a rich source of bioactive compounds, emphasizing its potential in scientific research and its expanding industrial applications in health, nutrition, and cosmetics.

## 1. Introduction

*Ulva lactuca*, commonly known as sea lettuce, has enormous chemical potential. This widespread green macroalgae is an important producer in marine ecosystems and thrives in coastal waters in temperate and tropical regions worldwide. However, the importance of *U. lactuca* goes far beyond its ecological role. Recent studies have focused on the remarkable adaptability of this alga, discovering numerous important chemical components used in various industries [1,2].

The key to unlocking the full potential of *U. lactuca* lies in understanding its complex chemical structure. This knowledge can pave the way for the development of new and sustainable applications [3]. Further research into the potential health-promoting components of *U. lactuca* would also be in the interest of the dietary supplement industry. It can also be used to fortify functional food formulations and supplement our diet by providing specific nutrients as well as bioactive ingredients such as vitamins, minerals, and polysaccharides [3,4,5]. Finally, the potential of *U. lactuca* as a source for biofuel production offers a glimpse into a future energy-efficient landscape. The seaweed may be used as a renewable feedstock for biofuel production, as it contains lipids and carbohydrates [6,7]. In addition, the chemical composition of *U. lactuca* could offer a wealth of opportunities for the pharmaceutical and cosmetic industries.

The high content of fatty acids and amino acids in *U. lactuca* from the Adriatic Sea also increases its potential to promote human health and sustainable practice [8]. Further research into the specific health benefits and optimal use of these compounds is essential to maximize the value of this versatile marine resource. *U. lactuca* contains a diverse range of fatty acids, which contribute to its nutritional value. Its fatty acid profile is predominantly composed of polyunsaturated fatty acids (PUFAs), including omega-3 and omega-6 fatty acids. Key fatty acids found in *U. lactuca* include *α*-linolenic acid (ALA), eicosapentaenoic acid (EPA), and docosahexaenoic acid (DHA) [9,10]. The presence of these essential fatty acids highlights *U. lactuca’s* role in supporting cardiovascular health and reducing inflammation. Its balanced fatty acid composition also makes it a valuable addition to a diet aimed at maintaining overall health and well-being.

The bright green color of *U. lactuca* is produced by a combination of pigments, with chlorophylls a and b being the most important for the absorption of sunlight and photosynthesis [2]. In addition to the chlorophylls, auxiliary pigments such as carotenoids absorb additional wavelengths of light and protect the alga from harmful sunlight. *U. lactuca* thrives in its marine habitat thanks to this mixture of pigments [11].

This study focuses specifically on *U. lactuca* harvested in the Adriatic Sea and highlights the unique chemical profile of this marine environment. To date, there has been no comprehensive characterization of the overall chemical composition of *U. lactuca* from this region. By thoroughly investigating the diverse chemical landscape of *U. lactuca* from the Adriatic Sea, this research aims to fill a significant gap in the existing literature. The focus is on the characterization of the biomolecules, revealing the astonishing diversity of nature’s chemical ingenuity. Fatty acids, volatile compounds, amino acids, and pigments are all analyzed. By thoroughly investigating the unique chemical composition of *U. lactuca* from the Adriatic Sea, this research lays a solid foundation for future study and development. This work will not only improve our understanding of this marine resource but also contribute to its meaningful utilization. Ultimately, discovering the secrets hidden in the chemical composition of *U. lactuca* has the potential to improve human diet and environmental well-being while contributing to the development of a more sustainable future.

## 2. Results and Discussion

### 2.1. The Headspace Composition Obtained by HS-SPME/GC-MS

The comparative studies on the odor characteristics of five seaweeds indicated that *Ulva* sp. received higher scores for seaweed, marine, and seafood odor [12]. *Ulva* sp. chemical composition was found to vary depending on geographical distribution and seasons; the principal environmental factors were water temperature, salinity, light, nutrients, and mineral availability [13,14,15], and therefore, it was of interest to investigate *U. lactuca* from the Adriatic Sea. Two fibers of different polarities (PDMS/DVB and DVB/CAR/PDMS) were used for collecting *U. lactuca*, and a total of 28 compounds were extracted by HS-SPME and identified by GC-MS with relatively different compound abundance among the fibers (Table 1).

Unsaturated alkane heptadec-8-ene ranged in the headspace from 21.80% to 12.46%. It was previously detected in the headspace of dehydrated *U. lactuca* from Spain [12], fresh *U. lactuca* from China [16], or the extract of *U. lactuca* from Egypt. The amount of heptadec-8-ene in brown and green algae is generally very low but was enriched when these algae were damaged mechanically or physiologically [17,18,19]. In comparison, the percentages of heptadec-8-ene in red algae *Pyropia yezoensis*, *Pyropia haitanensis*, and *Bangia* sp. (*Rhodophyta*) were 30–50% of the headspace volatile compounds, much higher than in the green alga *Ulva australis* (formerly *Ulva pertusa*) (*Chlorophyta*), the brown alga *Sargassum thunbergii*, and the red alga *Gracilariopsis lemaneiformis* (formerly *Gracilaria lemaneiformis*) [20]. From the marine green alga, *Bryopsis maxima* in a phosphate buffer (Z)-heptadec-8-ene was released by mechanical wounding, but this process was inhibited by heat treatment [17]. Therefore, it was supposed that (Z)-heptadec-8-ene would be also formed enzymatically from a fatty acid and might act as a chemical signal or a semiochemical. There is likely an enzymatic system (a type of heme protein) in *Pyropia* (*Rhodophyta*), which catalyzes eicosapentaenoic acid to produce heptadec-8-ene [20] with the optimal conditions of pH 9 and 25 °C, but the enzyme was thermal-resistant with over 50% activity remaining at 60–100 °C. Although the enzyme inhibitors phenidone, phenanthroline, and L-cysteine had no effect, the enzyme was significantly inhibited by hemoprotein NaN_3_, which is consistent with the result obtained for *Pyropia tenera* and *Pyropia* sp. (*Rhodophyta*) [20]. (*Z*)-heptadec-8-ene exhibited the reduction of reproduction of *Varroa destructor* [21].

Other abundant compounds were lower aldehydes heptanal (20.17%; 16.35%) and nonanal (10.57%; 6.62%), followed by minor percentages of pentanal (0.86%; 0.97%), hexanal (2.05%; 2.04%), octanal (1.86%; 1.11%), and decanal (2.12%; 1.13%). The algal aldehydes are formed through the degradation of fatty acids via the oxidation or enzymatic action of lipoxygenases [22]. Hexanal and heptanal are mainly derived from linoleic acid [23]. Nonanal could originate from *ω*-9 monounsaturated fatty acids (MUFAs) and *ω*-6 polyunsaturated fatty acids (PUFAs), such as linoleic acid [24].

Benzyl alcohol (8.53%; 9.71%) and benzaldehyde (2.61%; 2.61%) were the main benzene derivatives in the headspace. It was previously reported that benzyl alcohol was more abundant in the dry algal samples [25], while benzaldehyde could be decreased in the dried alga due to evaporation. The volatile benzene derivatives can be formed from phenylalanine when the side chain of a carbon skeleton shortens by C2-unit (via the oxidative pathway) [26]. In addition, benzaldehyde was readily reduced to benzyl alcohol by five cultures of photosynthetic microalgae [27].

The main volatile compounds in *U. lactuca* headspace from Spain were different: dimethyl sulfide, ethyl acetate, heptadecane, *α*-ionone, 2-phenylethyl acetate, and dimethyl sulfoxide [28]. Therefore, the obtained results agree with Narain [29] concerning a greater variety of volatile organic compounds identified in green seaweeds compared to red and brown species.

#### Volatiles Obtained by Hydrodistillation (HD)

The composition of hydrodistillate (Table 1) was considerably different in comparison to the headspace composition as was expected. Only five common compounds were found (heptanal, benzaldehyde, (*E*,*Z*)-hepta-2,4-dienal, (*E*,*E*)-hepta-2,4-dienal, 2,6-dimethylcyclohexanol, and heptadec-8-ene) but with considerably lower abundance in the hydrodistillate. A total of 30 compounds were identified in the hydrodistillate with higher aliphatic compounds as dominant.

The major compounds of the hydrodistillate were (Z,Z,Z)-hexadeca-7,10,13-trienal (22.67%), followed by hexadecanoic acid (13.18%), pentadecanal (10.61%), and methyl hexadeca-4,7,10,13-tetraenoate (7.50%). It was previously reported that (Z,Z,Z)-hexadeca-7,10,13-trienal is the main volatile compound in the green algae, a typical component of the essential oil of marine green alga *U. pertusa* [30,31], together with the major characteristic compounds pentadecanal, (*Z*)-heptadec-8-enal, (*Z*,*Z*)-heptadeca-8,11-dienal, and (Z,Z,Z)-heptadeca-8,11,14-trienal, also characteristic for *Ulvaceae* essential oils [32]. They accounted for ca. 40% of the hydrodistillate and were considered to exhibit the characteristic flavor of seaweeds [31]. These aldehydes were shown to be produced enzymatically from unsaturated fatty acids in *U. pertusa* collected from the sea [32]. In addition, the long-chain aldehydes formed by the enzymes from unsaturated fatty acids of the thalli culture were the same as in the field fronds, i.e., (*Z*)-heptadec-8-enal from oleic acid, (Z,Z)-heptadeca-8,11-dienal from linoleic acid, (*Z*,*Z*,*Z*)-heptadeca-8,11,14-trienal from *α*-linolenic acid, and (*Z*,*Z*,*Z*)-heptadeca-5,8,11-trienal from *γ*-linolenic acid [31]. (*Z*,*Z*,*Z*)-hexadeca-7,10,13-trienal was reported as antioxidant [33].

### 2.2. Fatty Acid Composition and Nutritional Indices of Ulva lactuca

The fatty acid composition of a freeze-dried sample of green macroalgae *U. lactuca* is presented in Table 2.

Palmitic acid (C16:0) was the predominant fatty acid (45.16%) and together with oleic acid isomers (C18:1n9c+t) and arachidic acid (C20:0) comprises more than 70% of the total identified fatty acids. The total saturated fatty acids (SFAs) were 73.35%, while unsaturated fatty acid (UFA) content was 27.53%, with similar content of monounsaturated and polyunsaturated fatty acids. High SFA content with a predominance of palmitic acid in *U. lactuca* was reported by Mohy El-Din [34] and Peñalver et al. [35]. Although present in lower amounts than SFAs, the content of essential fatty acids, cis-linoleic acid (C18:2n6c) and *α*-linolenic acid (C18:3n3), was notable (4.69 ± 0.62% and 5.82 ± 0.51%, respectively). Based on the results of nutritional indices, *U. lactuca* is characterized by a low PUFA/SFA ratio and a high index of atherogenicity (IA) and thrombogenicity (IT) that is slightly less favorable compared to other green macroalgae but comparable or even lower than nutritional indices of food products of animal origin (meat and dairy products) [36]. A similar PUFA/SFA ratio was determined in our previous study for another green macroalgae, *Codium adhaerens* [37]. n-6 and n-3 fatty acids are very important in human and fish nutrition, and a low n-6/n-3 ratio (preferably lower than 10) is beneficial to health. The n-6/n-3 ratio for *U. lactuca* obtained in this study was 0.99 which is comparable to or even slightly lower than the literature data [8,38]. Due to the favorable nutritional composition of fatty acids, incorporating *U. lactuca* consumption has significant advantages in a healthy diet and health preservation and reveals the possibility of using macroalgae as a source of bioactive components in functional food products.

### 2.3. Pigment Content of Ulva lactuca

The total quantified carotenoid content of *U. lactuca* was reported as 3.46 mg/100 g. Specifically, the concentrations of *β*-carotene, lutein, *α*-carotene, and *β*-cryptoxanthin were 1926.62 ± 0.36 μg/100 g, 1389.30 ± 0.60 μg/100 g, 131.81 ± 0.11 μg/100 g, and 12.05 ± 0.11 µg/100 g, respectively (Table 3). The results show remarkably high levels of *β*-carotene and lutein, which are consistent with the improvements observed in aquaculture environments as reported by Eismann et al. [39], who also found significant carotenoid concentrations in *U. lactuca*. Although the exact amounts differ, the observations confirm the presence of high amounts of lutein and *β*-carotene. Nevertheless, the content of *β*-cryptoxanthin was relatively low. Abd El-Baky et al. [40] reported that *U. lactuca* had the lowest concentration of *β*-cryptoxanthin, while the content of *β*-carotene was low compared to *β*-carotene in the present research. Abd El-Baqi et al. [40] reported the presence of 34 bioactive compounds in *U. lactuca*, including carotenoids such as *α*-carotene, all-trans *β*-carotene, 9-cis *β*-carotene, vioxanthin, astaxanthin, lutein, zeaxanthin, and cryptoxanthin.

The carotenoid composition of *U. lactuca* can vary due to factors such as environmental factors, growth, post-harvest storage, and season [41,42,43]. Recently, attention has focused on *Ulva* species rich in carotenoids with high neoxanthin or vioxanthin content, as these carotenoids increase free radical scavenging activity and oxidative stability. In addition, marine antioxidants are considered promising for the prevention of many modern diseases [44]. Antioxidant properties include quenching of singlet and triplet oxygen, scavenging of superoxide and hydroxyl radicals, and proven reducing power in phenols, carotenoids, terpenoids, and sulfated polysaccharides [45]. *β*-cryptoxanthin is considered a provitamin A, as it is converted into vitamin A (retinol) by the body. Similar to other carotenoids, *β*-cryptoxanthin acts as an antioxidant, potentially averting free radical damage to cells and DNA and helping to restore oxidative DNA damage [46].

### 2.4. Amino Acids in U. lactuca

The amino acid composition of macroalgae *U. lactuca* is presented in Table 4. Amino acid content was expressed as mg of amino acid per 100 g of proteins. The protein content in the *U. lactuca* freeze-dried sample was 17.66 ± 0.54%. The obtained protein content complied with results reported by Debbarma et al. [47] and Peñalver et al. [35], but lower and higher results are available in the literature [1,8,33,38]. The differences in protein content are attributed to different geographical origins, seasons, and environmental conditions (like sea temperature and salinity). The major amino acids determined in this study were glycine, threonine, aspartic, and glutamic acid (Table 4). *U. lactuca* contains all essential amino acids (EAA) in considerable amounts, except histidine and leucine content, which accounted for almost one-third of the total amino acid content. The content of threonine which accounted for about 50% of total EAA in *U. lactuca* was considerably higher compared to the threonine content in other edible algae (*Undaria pinnatifida*, *Arthrospira platensis*, *Himanthalia elongate*, and *Porphyra umbilicales*) from Spain where threonine content was about 1/10 of total EAA [35]. Threonine is an important amino acid for the formation of collagen, elastin, and tooth enamel and aids liver and lipotropic function when combined with aspartic acid and methionine. A low content of leucine was also found in sea lettuce from Spain [35], but in the literature, the limiting amino acids reported are also lysine, methionine, and isoleucine [8,35,38]. Partially, the difference can be attributed to the extraction procedure.

Namely, most of the studies perform acid hidrosis before derivatization, but according to Ummat et al. [48], the extraction using green solvents (water and citric acid) results in greater extraction yield, but additional parameters (temperature, time, and algae species) also have an impact on the amino acid profile. Aspartic and glutamic acid together with asparagine and glutamine contribute to the specific flavor and taste of the seaweeds, the umami taste. As presented in Table 4, aspartic and glutamic acid are present in large amounts in *U. lactuca* (123.77 ± 0.92 and 97.15 ± 1.70, respectively). Having in mind the obtained results, *U. lactuca* is a great source of proteins as well as amino acids that contribute to the specific umami taste and therefore can be added to enrich the taste of food products and possibly replace the umami additives that are increasingly used in food preparation.

### 2.5. Non-Volatile Compounds in Ethanol Extract

A non-targeted analysis of the ethanol extract from *U. lactuca* led to the tentative identification of 56 compounds. These include twenty-three derivatives of fatty acids, eleven phenolic acids, eight pigments and their derivatives, three flavonoids, and three steroids and their derivatives. Phenolic acids, flavonoids, and three fatty acid derivatives were found using ESI− mode, while the rest were found using ESI+ mode (Table 5).

Hexadecasphinganine (compound **11**) was identified as the most abundant compound (Table 5). It belongs to the complex class of lipids known as sphingolipids, whose spingoid bases serve as building blocks of cell membranes. They also play roles in intra- and extracellular signaling [49]. In marine organisms, they are found as secondary metabolites. Researchers have shown interest in them due to their antiviral, antibacterial, anti-inflammatory, antitumor, and immunostimulatory activities [50,51]. The presence of sphingosine derivatives has been reported in the ethanol extract of *Ulva lactuca (formerly Ulva fasciata)* [52,53], methanol extract of the sponge *Spheciospongia inconstans* (formerly *Spirastrella inconstans*) [54], and methanol–dichloromethane extracts of brown algae *Ericaria crinita* and *E. amentacea* [55].

Azelaic acid (compound **24**) was the second most abundant fatty acid derivative (Table 5). It has been recognized as an efficient therapeutic against some skin disorders such as acne and hyperpigmentation. It also has cytotoxic and antiproliferative activities in the case of human malignant melanocytes and has antibacterial properties [56]. The antibacterial behavior of diatom *Asterionellopsis glacialis* was probably due to the presence of azelaic acid [57]. Kalasariya et al. [58] detected it in methanolic extract of *Ulva lactuca* and proposed its antiviral activity against SARS-CoV-2 [58].

Among phenolics, 5-sulfosalicylic acid (compound **8**) was the most abundant, followed by 4-hydroxybenzoic acid-4-O-sulphate (compound **13**), vanillic acid 4-sulfate (compound **4**), and 4-hydroxybenzoic acid (compound **7**) (Table 5). They were all detected in [M-H]- mode. 5-sulfosalicylic acid, a derivative of salicylic acid that contains a sulfonate group, possesses antioxidant properties [59]. It has shown effectiveness against breast cancer cell lines (MCF-7) and HUVEC cells, with low toxicity [60]. 4-hydroxybenzoic acid-4-O-sulphate, also known as 4-(sulfooxy) benzoic acid, was found in the methanol extract of green alga *Dasycladus vermicularis* [61]. Welling et al. highlighted the importance of sulfated phenolic metabolites, which function as storage forms for more active metabolites [62]. Although limited data exist on vanillic acid 4-sulfate, this storage role may also apply to it. 4-hydroxybenzoic acid, which has antibacterial, antifungal, anti-inflammatory, and antioxidant properties, was the highest phenolic compound detected in methanol–hexane extract *U. fasciata* [63].

Pheophytin a (compound **47**), hydroxypheophytin a (compound **45**), and pheophorbide a (compound **29**) were the most abundant among pigment derivatives (Table 5). Chlorophylls and pheophytins are used in the food industry as antioxidants [64]. Pheophytin a [65] and pheophorbide a [66] detected in ethanol extract of *Ulva prolifera* (previously *Enteromorpha prolifera*) have been linked to antigenotoxic and antitumor-promoting activities and antioxidant properties, respectively.

## 3. Materials and Methods

### 3.1. Macroalga Sampling and Extraction

Macroalga *Ulva lactuca* was collected in May through a single-site sampling from the Adriatic Sea. Geographical coordinates were 44°07′00″ N, 15°14′00″ E (Zadar). The collection occurred at the sea depth of 1 m, with a sea temperature of 16 °C. The collected algae and seawater were sealed in an airtight plastic bag and promptly transported to the laboratory after collection. The sample was kept in the dark at 4 °C for 48 h before further analysis.

The *U. lactuca* sample was washed with tap water (5×) and with deionized water (2×), then cut into 5–10 mm slices. For the analysis of volatiles, the slices were air-dried for 7 days at room temperature in the shadow. For all other analysis, the sample was stored in an ultra-low freezer (CoolSafe 55-9 PRO, Labogene, Allerød, Denmark) for 24 h at a temperature of −60 °C. After freezing, the sample underwent freeze-drying under a high vacuum (0.13–0.55 hPa) for 24 h. Primary and secondary drying temperatures were set to −30 °C and 20 °C, respectively.

### 3.2. The Headspace Analysis

The air-dried sample (1 g) was put in a 10 mL headspace vial, sealed with a PTFE-silicon septum, and extracted by a manual holder (Supelco, Bellefonte, PA, USA) using DVB/CAR/PDMS solid-phase microextraction (SPME) fiber (Supelco, Bellefonte, PA, USA) and PDMS/DVB fiber (Supelco, Bellefonte, PA, USA). The fibers were conditioned according to the manufacturer’s instructions. Equilibration of the sample was carried out for 15 min at 60 °C, after which the sample was extracted for 45 min. Thermal desorption of the fiber was performed directly to the GC column for 6 min at 250 °C. Three independent extractions were conducted, and the obtained results were reported as mean values. Hydrodistillation (HD) was performed according to a previously published procedure by Radman et al. [37].

A gas chromatograph (7890B Agilent Technologies, Palo Alto, Santa Clara, CA, USA) tandem mass-spectrometer detector (model 5977A MSD, Agilent Technologies) was used to analyze VOCs isolated from *U. lactuca*. The separation of VOCs was underrun on an HP-5MS capillary column (30 m × 0.25 mm, 0.25 µm film thickness, 19091 S-433 UI-INT Agilent Technologies, Palo Alto, Santa Clara, CA, USA). The GC–MS analysis conditions and the identification procedure of the compounds were as specified by Jokić et al. [67].

The volatiles were identified by with their mass spectra comparing with the spectra from the NIST08 and Wiley275 libraries. The National Institute of Standards and Technology (NIST) Gas Chromatography Library (http://webbook.nist.gov/chemistry/, 10 June 2024) was used for the comparison with calculated retention indices (RIs) for each compound. RIs were calculated based on the retention times of C_9_-C_20_ *n*-alkanes under the same GC-MS working conditions.

### 3.3. Gas Chromatography Flame-Ionization Detection Analysis of Fatty Acids

Determination of fatty acids in a freeze-dried sample of *U. lactuca* was performed using the GC-FID method as described in our previous papers [37,68]. Briefly, macroalgae lipid fraction was extracted using the Folch method, after which the fatty acids were transesterified into volatile methyl esters (FAME) using methanolic potassium hydroxide. After filtration of separated FAMEs through a 0.45 μm membrane nylon filter, the sample was injected into the GC system.

### 3.4. Determination of Protein Content and Determination of Amino Acids Using HPLC-FLD Method

Protein content in *U. lactuca* was determined by the standard Kjeldahl method. Nitrogen content obtained by the Kjeldahl method was multiplied with a conversion factor (6.25) to calculate protein content in *U. lactuca*.

HPLC analysis was performed with the Shimadzu HPLC system consisting of a Shimadzu LC-20AD solvent delivery module, Shimadzu CTO-20AC column oven, Shimadzu SIL-10AF autosampler, and Shimadzu RF-20Axs fluorescence detector coupled with LabSolution Lite software (Release 5.52). The macroalgae sample (1 g) was dissolved in 25 mL of ultrapure water, vortexed, and placed into an ultrasonic bath for 15 min for incubation. After filtration through filter paper (21/N), an aliquot of filtrate (120 μL) was used for derivatization with OPA (*o*-phthalaldehyde) and FMOC (9-Fluorenylmethoxycarbonyl chloride). Filtrate aliquot was placed in a glass tube, 150 µL of borate buffer (pH 10.2) was added and vortexed, and after 30 s, 30 µL of OPA reagent, 30 µL of FMOC reagent, and 1920 µL of ultrapure water were added. After the addition of each reagent, the mixture was vortexed. Before HPLC analysis, solutions were filtered through a 0.45 µm nylon membrane filter.

Separation of fluorescent amino acid derivates was performed on the Inertsil ODS-3V column (250 × 4.6 mm, 5 μm). Column temperature was set at 40 °C. The mobile phase was composed of 40 mM NaH_2_PO_4_ (pH 7.8) (solvent A), while solvent B was a solution of acetonitrile, HPLC grade methanol, and ultrapure water (45:45:10, *v*:*v*:*v*). Mobile phase flow was 1 mL/min with gradient conditions as follows: starting percentage of solvent B was 15%, linear increase in solvent B up to 55% at 40 min, hold up to 45 min, and at 65 min, the initial conditions were achieved. The detection of separated amino acids was at 340 nm (excitation wavelength) and 450 nm (emission wavelength) except for proline, where wavelengths were 266 nm and 305 nm, respectively. Identification was achieved by comparing the retention time of each amino acid in the sample with the retention time of the same amino acid in a standard solution (AA standard 1 nmol/μL, Agilent Technologies and L-tryptophan, Acros Organics, Geel, Belgium). Quantification of identified components was performed with an external calibration method, and the results were expressed as mg of amino acid/100 g protein.

### 3.5. Determination of Pigments Using High-Performance Liquid Chromatography (HPLC-DAD) Method

Quantification of individual carotenoid compounds was performed by high-resolution liquid chromatography (High-Performance Liquid Chromatography, HPLC) on Shimadzu Prominence equipment (Shimadzu, Kyoto, Japan), which contains an LC-20AT binary pump, CTO-20A thermostat, and SIL -20A automatic dispenser connected to a DAD detector. The separation was performed on a column GRACE-Vydac 201TP54 C18, 250 4.6 mm, 5 mm (Hesperia, CA, USA). A solvent used as the mobile phase was stabilized methanol: stabilized THF (95:5, *v*/*v*), flow rate 1 mL/min. Samples and solvents were filtered before analysis through 0.45 µm pore size membrane filters (Millipore, Bedford, MA, USA). Carotenoids were detected at 445 nm. The wavelength range used was 200–600 nm.

### 3.6. Ultra-High-Performance Liquid Chromatography–High-Resolution Mass Spectrometry (UHPLC-ESI-HRMS) of Ethanol Extract

A freeze-dried sample weighing 1 g was extracted with ethanol under sonication for 60 min at 40 °C. The sample was filtered through a 0.2 µm PVDF syringe filter (Agilent Technologies). The UHPLC-ESI-HRMS analyses were conducted with an ExionLC AD UHPLC system (AB Sciex, Concord, ON, Canada), coupled to a TripleTOF 6600+ quadrupole time-of-flight (Q-TOF) mass spectrometer (AB Sciex, Concord, ON, Canada) featuring a duospray ion source. Electrospray ionization was operated in positive (ESI+) and negative (ESI−) modes using collision-induced dissociation (CID) in information-dependent acquisition (IDA) mode to collect MS/MS mass spectra. Gases were set as follows: nebulizer gas (GS1) 40, heater gas (GS2) 15, and curtain gas (CUR) 30. The temperature for positive mode was set to 500 °C, while negative mode was 400 °C. Ion spray voltage was set at 5000 V (ESI+) and −4000 V (ESI−). Declustering potential (DP) was set to 80 V (ESI+) and −35 V (ESI−), while collision energy (CE) was set to 40 V (ESI+) and 30 V (ESI−) with the 20 V collision energy spread. ACD/Spectrus Processor 2021.1.0 software (ACD/Labs, Toronto, ON, Canada) was used for data processing. Compound identification was proposed based on mass spectra, reported elemental compositions, and database searches in MassBank, Lipid Maps, ChemSpider, and ChEBI.

Chromatographic separation of the polyphenolic compounds (ESI−) was achieved with an Acquity UPLC CSH Phenyl-hexyl analytical column (Waters, Milford, MA, USA), while for the compounds in ESI+, separation was achieved with an Acquity UPLC CSH Phenyl-Hexyl analytical column (Waters, Milford, MA, USA), both measuring 2.1 mm × 100 mm and featuring a particle size of 1.7 µm. The mobile phases consisted of water with 0.1% formic acid (phase A) and acetonitrile with 0.1% formic acid (phase B). The analysis was carried out at a steady flow rate of 0.4 mL/min, with the column oven maintained at 30 °C. The injection volume was kept at 4 µL. Gradient elution was set as described in Table 6.

## 4. Conclusions

This extensive chemical profiling of *Ulva lactuca* from the Adriatic Sea revealed the presence of various compounds, some of which have bioactive properties. HS-SPME/GC-MS and analysis of the hydrodistillate revealed the presence of various volatile organic compounds such as various aldehydes, benzyl alcohol, (*Z*,*Z*,*Z*)-hexadeca-7,10,13-trienal, and hexadecanoic acid, all of which are used in the cosmetic industry. The predominance of saturated fatty acids, especially palmitic acid, together with a favorable ratio of n-6/n-3 polyunsaturated fatty acids, makes this alga a valuable component of a balanced diet. The analysis of amino acids revealed that it is an excellent source of proteins and amino acids such as leucine, valine, and isoleucine, which play an important role in muscle repair and metabolic health. The presence of carotenoids, especially β-carotene and lutein, enhances the value of this algae as a source of antioxidants. In addition, 56 compounds were identified when analyzing the ethanol extract, including derivatives of fatty acids, phenolic acids, pigments, flavonoids, and steroids. In particular, several compounds such as hexadecasphinganine; azelaic acid; phenolics like 5-sulfosalicylic acid, 4-hydroxybenzoic acid 4-O-sulphate, vanillic acid 4-sulphate, and 4-hydroxybenzoic acid; and pigment derivatives such as pheophytin *a*, hydroxypheophytin *a*, and pheophorbide *a* have shown various functions or potential health benefits, including antiviral, antibacterial, anti-inflammatory, antitumor, and antioxidant activities. Given its rich chemical composition and the proven presence of numerous bioactive compounds, *Ulva lactuca* holds significant potential for use in various industries, including environmental protection, food production, pharmaceuticals, and cosmetics.

## Figures and Tables

**Table 1 ijms-25-11711-t001:** The volatile compounds from *Ulva lactuca* isolated by headspace solid-phase microextraction (HS-SPME) and analyzed by gas chromatography-mass spectrometry (GC-MS): *U. lactuca* extracted by DVB/CAR/PDMS fiber and by PDMS/DVB fiber. HD, hydrodistillation.

No.	Compound	Rt	RI	PDMS/DVB	DVB/CAR/PDMS	HD
1.	Dimethyl sulfide	1.565	<900	4.46	5.81	-
2.	(*E*)-But-2-enal	1.837	<900	0.05	3.48	-
3.	Pent-1-en-3-ol	1.943	<900	2.10	3.48	-
4.	Pentanal	2.001	<900	0.86	0.97	-
5.	3-Methylbut-2-enal	2.364	<900	0.71	1.54	-
6.	Hexanal	2.734	<900	2.05	2.04	-
7.	(*E*)-Hex-2-enal	3.387	<900	0.49	0.31	-
8.	Hexan-1-ol	3.549	<900	1.40	1.11	-
9.	Heptanal	4.150	904	20.17	16.35	0.33
10.	(*E*)-Hept-2-enal	5.343	961	0.84	1.23	-
11.	Benzaldehyde	5.527	968	2.61	2.61	0.44
12.	2-Pentylfuran	6.197	992	-	-	0.24
13.	(*E*,*Z*)-Hepta-2,4-dienal	6.410	999	0.11	0.35	1.45
14.	Octanal	6.552	1004	1.86	1.11	-
15.	2-(2-Ethoxyethoxy)- ethanol (Carbitol)	6.621	1007	-	0.70	-
16.	(*E*,*E*)-Hepta-2,4-dienal	6.817	1014	3.73	1.89	0.50
17.	Benzyl Alcohol	7.663	1043	8.53	9.71	-
18.	(*E*)-Oct-2-enal	8.291	1062	2.98	5.56	-
19.	Heptanoic acid	9.215	1087	0.05	0.74	-
20.	(*E*,*E*)-Octa-3,5-dien-2-one	9.560	1095	0.11	0.66	-
21.	Nonanal	9.914	1105	10.57	6.62	-
22.	2,6-Dimethylcyclohexanol	10.119	1111	1.58	0.76	0.18
23.	(*E*,*Z*)-Nona-2,6-dienal	11.821	1157	0.22	0.80	-
24.	(*E*)-Non-2-enal	12.046	1163	1.69	1.80	-
25.	Decanal	13.900	1206	2.12	1.13	-
26.	(*E*)-Dec-2-enal	16.225	1265	0.05	0.54	-
27.	(*E*,*Z*)-Deca-2,4-dienal	17.578	1294	-	-	0.29
28.	(*E*,*E*)-Deca-2,4-dienal	18.529	1318	-	-	3.15
29.	α-Ionone	23.091	1429	-	-	0.20
30.	(*E*)-6,10-Dimethylundeca-5,9-dien-2-one	24.118	1455	-	-	0.15
31.	Dodecan-1-ol	25.100	1478	-	-	1.01
32.	trans-β-Ionone	25.438	1486	-	-	3.04
33.	Pentadec-1-ene	25.688	1492	0.11	0.56	-
34.	Pentadecane	26.003	1500	2.69	4.24	-
35.	Tridecanal	26.428	1511	-	-	0.28
36.	Tridecan-1-ol	29.068	1580	-	-	0.23
37.	Tetradecanal	30.352	1613	-	-	0.99
38.	Heptadec-8-ene	32.764	1679	21.80	12.46	4.39
39.	(*Z*)-Pentadec-11-enal	33.333	1693	-	-	1.66
40.	Heptadecane	33.562	1700	-	0.49	-
41.	Pentadecanal	34.098	1715	-	-	10.61
42.	Tetradecanoic acid	36.186	1775	-	-	3.93
43.	(*Z*)-Hexadec-11-enal	36.938	1795	-	-	0.39
44.	Hexahydrofarnesyl acetone	38.637	1846	-	-	0.79
45.	Hexadecan-1-ol	38.839	1852	-	-	2.29
46.	(Z,Z,Z)-Hexadeca-7,10,13-trienal	39.097	1863	-	-	1.61
47.	(*Z*)-Hexadec-9-en-1-ol	39.208	1863	-	-	1.12
48.	Methyl hexadeca-4,7,10,13-tetraenoate	39.757	1879	-	-	7.50
49.	Hexadecan-1-ol	39.913	1884	-	-	1.34
50.	(*Z*,*Z*,*Z*)-Hexadeca-7,10,13-trienal	40.237	1893	-	-	22.67
51.	Dibutyl phthalate	42.483	1963	-	-	1.13
52.	Hexadecanoic acid	42.982	1978	-	-	13.18
53.	Sulfur, mol. (S8)	44.098	2012	1.16	3.45	-
54.	Phytol	47.439	2114	-	-	4.15

RI—retention index; Rt—retention time.

**Table 2 ijms-25-11711-t002:** Fatty acid composition determined by GC-FID and nutritional indices of *Ulva lactuca*.

No.	Fatty Acid	Av ± SD (%)
1.	Dodecanoic acid (lauric acid) (C12:0)	4.43 ± 0.19
2.	Tetradecanoic acid (myristic acid) (C14:0)	2.76 ± 0.58
3.	Hexadecanoic acid (palmitic acid) (C16:0)	45.16 ± 1.82
4.	Octadecanoic acid (stearic acid) (C18:0)	5.40 ± 0.55
5.	Eicosanoic acid (arachidic acid) (C20:0)	14.57 ± 0.84
6.	Docosanoic acid (behenic acid) (C22:0)	1.03 ± 0.08
Total saturated fatty acids (SFAs)		73.35
7.	Palmitoleic acid (C16:1)	2.37 ± 0.17
8.	Cis-oleic acid+trans-oleic acid (C18:1n9c+t)	13.59 ± 1.47
Total monounsaturated fatty acids (MUFAs)		15.96
9.	Cis-linoleic acid (C18:2n6c)	4.69 ± 0.62
10.	α-linolenic acid (C18:3n3)	5.82 ± 0.51
11.	Docosadienoic acid (C22:2n6)	1.06 ± 0.11
Total polyunsaturated fatty acids (PUFAs)		11.57
Total n-3 fatty acids (n-3 PUFAs)		5.82
Total n-6 fatty acids (n-6 PUFAs)		5.75
Nutritional indices		
PUFA/SFA		0.16
Index of atherogenicity (IA)		2.20
Index of thrombogenicity (IT)		1.82
Hypocholesterolemic/hypercholesterolemic ratio (HH)		0.48
Unsaturation index (UI)		44.92

Av—average value of three replicates expressed in percentage (%) with standard deviation (SD).

**Table 3 ijms-25-11711-t003:** Carotenoid content in *Ulva lactuca*.

Carotenoid	Content (µg/100 g)
*β*-carotene	1926.62 ± 0.36
lutein	1389.30 ± 0.60
*α*-carotene	131.81 ± 0.11
*β*-cryptoxanthin	12.05 ± 0.11

**Table 4 ijms-25-11711-t004:** Amino acid composition of *Ulva lactuca* determined by HPLC-FLD.

Amino Acid	Av ± SD (mg/100 g Protein)
Aspartic acid	123.77 ± 0.92
Glutamic acid	97.15 ± 1.70
Serine	32.73 ± 0.15
Histidine	3.19 ± 0.02
Glycine	182.15 ± 1.59
Threonine	135.11 ± 1.68
Arginine	50.76 ± 4.03
Alanine	53.80 ± 0.73
Tyrosine	9.60 ± 0.09
Cystine	56.98 ± 1.44
Valine	14.73 ± 0.25
Methionine	16.04 ± 0.12
Tryptophane	15.41 ± 0.63
Phenylalanine	21.86 ± 0.93
Isoleucine	12.73 ± 0.21
Leucine	9.85 ± 0.35
Lysine	38.04 ± 1.99
Proline	50.23 ± 3.47

Av—average value of duplicates expressed as mg amino acid/100 g protein with standard deviation (SD).

**Table 5 ijms-25-11711-t005:** Major non-volatile compounds in *U. lactuca* ethanol extract identified using high-performance liquid chromatography–high-resolution mass spectrometry with electrospray ionization (UHPLC-ESI–HRMS) in both positive (ESI+) and negative (ESI−) mode.

No.	Name	Mass	[M-H]^−^ or [M+H]^+^	Molecular Formula	t_R_ (min)	Mass Difference (ppm)	Peak Area (Arbitrary Units)
Phenolic acids							
3	4-Hydroxybenzaldehyde	122.037	121.02950	C_7_H_6_O_2_	1.55	0.0	1.42 × 10^4^
7	4-Hydroxybenzoic acid	138.032	137.02442	C_7_H_6_O_3_	5.668	7.2	2.89 × 10^5^
35	3-Hydroxyphenylacetic acid	152.047	151.04007	C_1_H_1_O_3_	16.554	0.0	2.46 × 10^3^
18	Hydroxytyrosol	154.063	153.05572	C_1_H_10_O_3_	11.821	1.7	4.15 × 10^4^
48	5-(3,4-Dihydroxyphenyl)pentanoic acid	210.089	209.08193	C_11_H_14_O_4_	20.117	7.2	3.35 × 10^4^
13	4-Hydroxybenzoic acid-4-O-sulphate	217.989	216.98123	C_7_H_6_O_6_S	8.601	6.6	7.30 × 10^5^
8	5-Sulfosalicylic acid	217.989	216.98123	C_7_H_6_O_6_S	5.684	6.8	3.35 × 10^6^
4	Vanillic acid 4-sulfate	247.999	246.99180	C_1_H_1_O_7_S	3.194	7.2	4.21 × 10^5^
2	Caffeic acid 4-O-sulfate	259.999	258.99180	C_9_H_1_O_7_S	1.414	3.3	2.30 × 10^4^
5	4-(β-D-Glucosyloxy)benzoic acid	300.085	299.07724	C_13_H_16_O_1_	3.804	0.7	2.17 × 10^3^
10	Caffeic acid 4-O-glucuronide	356.074	355.06707	C_1_5H_16_O_10_	7.091	9.8	6.86 × 10^3^
Flavonoids							
1	Sativanone	300.1	299.09250	C_17_H_16_O_5_	0.872	9.7	1.02 × 10^5^
6	3-Hydroxyterphenyllin	354.11	353.10306	C_20_H_18_O_6_	4.193	6.2	8.76 × 10^4^
15	3,3′′-Dihydroxyterphenyllin	370.105	369.09798	C_20_H_18_O_7_	10.872	1.6	5.07 × 10^4^
Fatty acid derivatives							
16	Ethyl 3-oxohexanoate	158.094	157.08702	C_8_H_14_O_3_	10.872	4.3	3.28 × 10^6^
17	Ethyl 2-ethylacetoacetate	158.094	157.08702	C_8_H_14_O_3_	11.177	3.5	1.43 × 10^6^
24	Azelaic acid	188.105	187.09758	C_9_H_16_O_4_	14.296	5.4	4.09 × 10^6^
9	Loliolide	196.11	197.11722	C_11_H_16_O_3_	6.227	2.3	3.32 × 10^6^
19	Tetradecanamide	227.225	228.23219	C_14_H_29_NO	12.492	11.6	7.57 × 10^4^
12	Palmitoleamide	253.241	254.24784	C_16_H_31_NO	8.157	0.7	5.25 × 10^4^
21	Palmitamide	255.256	256.26349	C_16_H_33_NO	13.688	2.5	5.27 × 10^5^
11	Hexadecasphinganine	273.267	274.27406	C_16_H_35_NO_2_	7.903	0.4	1.53 × 10^8^
20	Linoleamide	279.256	280.26349	C_18_H_33_NO	13.415	2.2	6.35 × 10^5^
23	9-Octadecenamide	281.272	282.27914	C_18_H_35_NO	14.081	0.4	1.11 × 10^6^
26	Octadecanamide	283.288	284.29479	C_18_H_37_NO	14.781	6.1	2.63 × 10^5^
14	Palmitoylethanolamide	299.282	300.28971	C_18_H_37_NO_2_	10.76	0.0	1.40 × 10^4^
28	11-Eicosenamide	309.303	310.31044	C_20_H_39_NO	15.054	4.6	2.16 × 10^5^
33	Erucamide	337.334	338.34174	C_22_H_43_NO	15.961	0.3	3.01 × 10^6^
22	Glycerol palmitate	330.277	331.28429	C_19_H_38_O_4_	13.979	6.8	8.23 × 10^4^
27	Glycerol monostearate	358.308	359.31559	C_21_H_42_O_4_	15.02	4.1	1.95 × 10^5^
34	1-(9Z-octadecenoyl)-2-(9Z-tetradecenoyl)-glycero-3-phosphocholine	729.531	730.53813	C_40_H_76_NO_8_P	16.388	7.6	9.24 × 10^3^
36	1-(9Z-octadecenoyl)-2-(9Z-pentadecenoyl)-glycero-3-phosphocholine	743.547	744.55378	C_41_H_78_NO_8_P	16.627	7.8	7.72 × 10^3^
42	1-(9Z-octadecenoyl)-2-(9Z-nonadecenoyl)-glycero-3-phosphocholine	799.609	800.61638	C_45_H_86_NO_8_P	18.425	3.3	5.21 × 10^3^
41	1-(11Z,14Z-eicosadienoyl)-2-heptadecanoyl-glycero-3-phosphoserine	801.552	802.55926	C_43_H_80_NO_10_P	18.066	6.0	3.33 × 10^4^
43	1-(11Z,14Z-eicosadienoyl)-2-nonadecanoyl-glycero-3-phosphoserine	829.583	830.59056	C_45_H_84_NO_10_P	19.161	1.3	6.87 × 10^3^
37	3-{[6-O-(α-D-Galactopyranosyl)-β-D-galactopyranosyl]oxy}-2-[(9Z)-9-hexadecenoyloxy]propyl (9Z,12Z,15Z)-9,12,15-octadecatrienoate	912.581	913.58830	C_49_H_84_O_15_	16.953	0.0	3.69 × 10^2^
38	1-hexadecanoyl-2-(9Z,12Z,15Z-octadecatrienoyl)-3-O-(α-D-galactosyl-1-6-β-D-galactosyl)-sn-glycerol	914.597	915.60395	C_49_H_86_O_15_	17.021	5.0	5.67 × 10^3^
Pigments and derivatives							
31	(2E)-3-[21-(Methoxycarbonyl)-4,8,13,18-tetramethyl-20-oxo-9,14-divinyl-3,4-didehydro-3-24,25-dihydrophorbinyl]acrylic acid	586.222	587.22890	C_35_H_30_N_4_O_5_	15.499	1.4	3.87 × 10^3^
30	3-[(21R)-21-(Methoxycarbonyl)-4,8,13,18-tetramethyl-20-oxo-9,14-divinyl-3,4-didehydro-3--24,25-dihydrophorbinyl]propanoic acid	588.237	589.24455	C_35_H_32_N_4_O_5_	15.465	2.9	7.37 × 10^3^
29	Pheophorbide a	592.269	593.27585	C_35_H_36_N_4_O_5_	15.362	2.5	2.95 × 10^4^
25	Fucoxanthin	658.423	659.43062	C_42_H_58_O_6_	14.747	4.5	6.10 × 10^3^
44	Divinyl pheophytin a	868.55	869.55755	C_55_H_72_N_4_O_5_	19.949	0.0	1.21 × 10^3^
47	Pheophytin a	870.566	871.57320	C_55_H_74_N_4_O_5_	20.103	1.0	2.87 × 10^5^
45	3-Phorbinepropanoic acid, 9-acetyl-14-ethylidene-13,14-dihydro-21-(methoxycarbonyl)-4,8,13,18-tetramethyl-20-oxo-, 3,7,11,15-tetramethyl-2-hexadecen-1-yl ester	886.561	887.56811	C_55_H_74_N_4_O_6_	19.949	5.1	7.19 × 10^4^
46	Methyl (3R,10Z,14Z,20Z,22S,23S)-12-ethyl-3-hydroxy-13,18,22,27-tetramethyl-5-oxo-23-(3-oxo-3-{[(2E,7R,11R)-3,7,11,15-tetramethyl-2-hexadecen-1-yl]oxy}propyl)-17-vinyl-4-oxa-8,24,25,26-tetraazahexacycl;o[19.2.1.16,9.111,14.116,19.02,7]heptacosa-1(24),2(7),6(27),8,10,12,14,16,18,20-decaene-3-carboxylate	902.556	903.56303	C_55_H_74_N_4_O_7_	19.966	4.2	2.11 × 10^4^
Steroids and derivatives							
40	(3β,20*R*,22*E*,24*S*)-Stigmasta-5,22-dien-3-ol	394.36	395.36723	C_29_H_46_	17.518	6.6	5.99 × 10^4^
32	7-Dehydrocholesteryl acetate	426.35	427.35706	C_29_H_46_O_2_	15.739	3.8	5.66 × 10^4^
39	(3β)-3-Hydroxystigmast-5-en-7-one	428.365	429.37271	C_29_H_4_8O_2_	17.381	2.0	9.12 × 10^3^

**Table 6 ijms-25-11711-t006:** Gradient elution for ESI+ (Acquity UPLC CSH Phenyl-Hexyl) and ESI− mode (Acquity UPLC BEH C8).

ESI+			ESI−		
Time (min)	A (%)	B (%)	Time (min)	A (%)	B (%)
0.0	98	2	0.0	100	0
0.6	98	2	2.0	95	5
18.5	0	100	25.0	55	45
25.0	0	100	30.0	0	100
			35.0	0	100

Note: starting conditions were set in 0.1 min, and column equilibration was achieved in 2 min.

## Data Availability

Data are available from authors for a limited time.

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
