# Peer review of "Comprehensive Phytochemical Profiling of Ulva lactuca from the Adriatic Sea"

_ijms, 2024, doi:10.3390/ijms252111711_

Round 1
Reviewer 1 Report
Comments and Suggestions for Authors
The manuscript entitled "Comprehensive Phytochemical Profiling of Ulva lactuca from the Adriatic Sea" addresses a topic relevant and appropriate for this journal.
Overall, the manuscript is well written and well reasoned.
However, the authors will have to introduce several corrections, especially in terms of the taxonomy of the organisms mentioned, as I indicate below.
It should be noted that in Science, authors always have to use valid names and not synonyms.
Corrections needed:
line 60 - fatty acids highlights U. lactuca's role in supporting cardiovascular health and reducing (Note: "'s" of lactuca is not in italics)
line 97 - heptadec-8-ene in red algae Pyropia yezoensis, Pyropia haitanensis, and Bangia sp. (Rhodophyta) were 30–
line 98 - 50% of the headspace volatile compounds, much higher than in the green alga Ulva australis (formerly Ulva pertusa) (Chlorphyta),
line 99 - the brown alga Sargassum thunbergii and the red alga Gracilariopsis lemaneiformis (formerly Gracilaria lemaneiformis)[20]. From
line 104 - enzymatic system (a type of heme protein) in Pyropia (Rhodophyta), which catalyzes eicosapentaenoic
line 108 - sult obtained for Pyropia tenera and Pyropia sp. (Rhodophyta) [20].
line 170/171 - PUFA/SFA ratio was determined in our previous study for another green macroalgae Codium adhaerens [35].
line 244 - sphingosine derivatives has been reported in the ethanol extract of Ulva lactuca (formerly Ulva fasciata) (Chlorophyta) [51,52],
line 245 - methanol extract of the sponge Spheciospongia inconstans (formerly Spirastrella inconstans (Porifera) [53], and methanol:dichloromethane
line 256 - azelaic acid [56]. Kalasariya et al. [57] detected it in methanolic extract of Ulva lactuca (formerly U. fasciata) and
line 270 - hexane extract Ulva lactuca (formerly U. fasciata) [62].
line 274 - a [64] and pheophorbide a [65] detected in ethanol extract of Ulva prolifera (previously En-
line 330 - trapure water, vortexed, and placed into an ultrasonic bath for 15 min for incubation.
Author Response
On behalf of all of the co-authors, I would like to thank all of the Reviewers for their careful consideration of this manuscript and also for their thoughtful and constructive remarks. We have addressed the Reviewers comments and concerns in the revised manuscript.
Reviewer 1
The manuscript entitled "Comprehensive Phytochemical Profiling of Ulva lactuca from the Adriatic Sea" addresses a topic relevant and appropriate for this journal. Overall, the manuscript is well written and well-reasoned. However, the authors will have to introduce several corrections, especially in terms of the taxonomy of the organisms mentioned, as I indicate below. It should be noted that in Science, authors always have to use valid names and not synonyms.
Corrections needed:
Question 1: line 60 - fatty acids highlights U. lactuca's role in supporting cardiovascular health and reducing (Note: "s" of lactuca is not in italics)
Answer 1: Line 60: "s" corrected
Question 2:
line 97 - heptadec-8-ene in red algae Pyropia yezoensis, Pyropia haitanensis, and Bangia sp. (Rhodophyta) were 30–
line 98 - 50% of the headspace volatile compounds, much higher than in the green alga Ulva australis (formerly Ulva pertusa) (Chlorphyta),
line 99 - the brown alga Sargassum thunbergii and the red alga Gracilariopsis lemaneiformis (formerly Gracilaria lemaneiformis)[20]. From
line 104 - enzymatic system (a type of heme protein) in Pyropia (Rhodophyta), which catalyzes eicosapentaenoic
line 108 - sult obtained for Pyropia tenera and Pyropia sp. (Rhodophyta) [20].
Answer 2: Line 97-98: added "(Rhodophyta)"
Line 99: added "Ulva australis (formerly Ulva" and "(Chlorophyta)"
Line 100: added "Gracilariopsis lemaneiformis (formerly"
Line 105: added "(Rhodophyta)"
Line 110: added "Pyropia" and added "(Rhodophyta)"
Question 3:
line 170/171 - PUFA/SFA ratio was determined in our previous study for another green macroalgae Codium adhaerens [35].
Answer 3: Line 172: added Codium
Question 4:
line 244 - sphingosine derivatives has been reported in the ethanol extract of Ulva lactuca (formerly Ulva fasciata) (Chlorophyta) [51,52],
line 245 - methanol extract of the sponge Spheciospongia inconstans (formerly Spirastrella inconstans (Porifera) [53], and methanol:dichloromethane
Answer 4: The suggestions have been accepted. Suggestions have been added in lines 245-246.
Question 5:
line 256 - azelaic acid [56]. Kalasariya et al. [57] detected it in methanolic extract of Ulva lactuca (formerly U. fasciata) and
Answer 5: Line 257: deleted "fasciata" and added "lactuca"
Question 6:
line 270 - hexane extract Ulva lactuca (formerly U. fasciata) [62].
Answer 6: Line 271: deleted "fasciata" and added "lactuca"
Question 7:
line 274 - a [64] and pheophorbide a [65] detected in ethanol extract of Ulva prolifera (previously En-
Answer 7: Line 275: added "Ulva"
Question 8:
line 330 - trapure water, vortexed, and placed into an ultrasonic bath for 15 min for incubation.
Answer 8: Line 331: "minutes" replaced with "min"

Reviewer 2 Report
Comments and Suggestions for Authors
Many points need to improve
1- in the introduction, are there any toxic compounds found in U. lactuca alga, or the problems present due to the absorption of pollution
2- What sources identify U. lactuca in material and methods?
3-Please add the calculation of protein
4- Carotenoids were detected at 445 nm, How the separation of contents, B, carotene, alpha-carotene, and lutein
5- The percentage of lipids, pigments, and protein related to algal dry weight
6- Table identify IR
7- U. lactuca extracted by DVB/CAR/PDMS fiber and by PDMS/DVB fiber, HD hydrodistillation, please clear, which methods are significant to extract Ulva lactuca
Author Response
Many points need to improve:
Question 1:
1 – in the introduction, are there any toxic compounds found in U. lactuca alga, or the problems present due to the absorption of pollution
Answer 1: Ulva lactuca is generally considered as a safe and nutritious seaweed, commonly used as food or feed. However, it can absorb pollutants from its environment, which may introduce toxic compounds depending on the habitat. This potential contamination poses risks when Ulva lactuca is harvested from polluted waters, as it can accumulate heavy metals, such as lead, mercury, cadmium, and arsenic. These metals are harmful to human health if ingested in high amounts.
Additionally, Ulva lactuca can accumulate other environmental pollutants like polycyclic aromatic hydrocarbons (PAHs) and persistent organic pollutants (POPs), which are known for their toxicity. This makes the source of the algae critical for safe consumption or application in various industries.
If Ulva lactuca is grown in clean waters, it typically does not present toxicity issues. You might also highlight the importance of proper monitoring of the growth environment to avoid contamination.
Question 2:
2 – What sources identify U. lactuca in material and methods?
Answer 2: Marine botanist identified the alga Ulva lactuca based on its distinctive morphological characteristics, such as thallus shape and cell arrangement.
Question 3:
3 – Please add the calculation of protein
Answer 3: Determination of protein content was performed using Kjeldahl method, and standard equations are used for calculation of nitrogen content. Multiplication of the obtained nitrogen content with the conversion factor, the proportion of protein is calculated. The calculation of protein was added.
Question 4:
4 – Carotenoids were detected at 445 nm. How the separation of contents, B, carotene, alpha-carotene, and lutein?
Answer 4: The wavelength range used was 200–600 nm. The quantification of the peaks was performed with the external standard method by preparing the calibration curve for each carotenoids and were detected at 445 nm.
Question 5:
5 – The percentage of lipids, pigments, and protein related to algal dry weight
Answer 5: Ulva lactuca sample was lyophilized prior the analyses to in order to be in a stable state until the analyses were carried out. The water content/dry weight was not determined in lyophilized sample since the moisture in lyophilized sample is usually very low and the it does not significantly affect the presentation of results on dry matter or on the sample.
Question 6:
6 – Table identify RI
Answer 6: In Table 1 RI was supplemented with explanation „retention index“.
Question 7:
7 – U. lactuca extracted by DVB/CAR/PDMS fiber and by PDMS/DVB fiber, HD hydrodistillation, please clear, which methods are significant to extract Ulva Lactuca
Answer 7:
Both methods are important for characterization of U. lactuca volatiles and they are complementary. HS-SPME with both fibers (DVB/CAR/PDMS and PDMS/DVB) enabled the extraction and further GC-MS analysis if the most volatile headspace compounds. HD enabled isolation of volatile and semi-volatiles compounds that were further identified by GC-MS. In that way, the complete profile of headspace, volatile and semi-volatile compounds is determined for U. lactuca.

Reviewer 3 Report
Comments and Suggestions for Authors
1. Further discussion of the physiological functions and potential health effects of unsaturated alkanes and aldehydes is suggested in section 2.1 to enrich the discussion.
2. It is suggested that the amino acid composition of U. lactuca be compared with other common plants or algae to highlight its uniqueness and advantages.
3. In the discussion of volatile compounds, the range of heptadec-8-ene content was mentioned, but there was no explanation as to why the content of this compound varies. It is suggested that an explanatory paragraph could be added to discuss how different growth conditions or environmental factors affect the production and content of heptadec-8-ene.
4. Two fibers of different polarity were used in the paper to collect U. lactuca. what was the basis for the choice of these two specific fibers? Is it possible that other important compounds were missed?
5. The text mentions that the seaweed Ulva lactuca was collected in the Adriatic Sea in May by single-point sampling. Does the sampling time cover the key stages in the growth cycle of the seaweed? Does it take into account the effect of seasonal variations on the chemical composition of the seaweed?
6. It is mentioned that U. lactuca contains all essential amino acids except histidine and leucine, which account for almost one-third of the total amino acid content. Does the absence of histidine and leucine limit the potential of U. lactuca as a protein source? Are there ways to supplement these amino acids?
Author Response
Question 1:
- Further discussion of the physiological functions and potential health effects of unsaturated alkanes and aldehydes is suggested in section 2.1 to enrich the discussion.
Answer 1: It is difficult to mention exact physiological functions and potential health effects of unsaturated alkanes and aldehydes since they usually serve as chemical communication molecules usually as semiochemicals. However, to enrich the discussion, we added the specific activity of (Z)-heptadec-8-ene that reduces the reproduction of Varroa destructor (21) in brood cells as well as the activity of (Z,Z,Z)-Hexadeca-7,10,13-trienal as antioxidant (33).
- Milani, N.; Della Vedova, G:, Nazzi, F. (Z)-8-Heptadecene reduces the reproduction of Varroa destructorin brood cells, Apidologie2004, 35, 265-273
- Rautela, I.; Parveen, A.; Singh, P.: Sharma, M.D. GC-MS analyses of ethanolic leaf extract of medicinal plant Solanum nigrum, World J. Pharm Res. 2019, 8,2299-307.
Question 2:
- It is suggested that the amino acid composition of U. lactucabe compared with other common plants or algae to highlight its uniqueness and advantages.
Answer 2: It is very difficult to compare the values of amino acids with other reported results due to the different extraction procedure and methodology but we have compared the profile and percentages of specific amino acids in EAA and total EE to the other edible algae. The discussion was broadened with example.
Question 3:
- In the discussion of volatile compounds, the range of heptadec-8-ene content was mentioned, but there was no explanation as to why the content of this compound varies. It is suggested that an explanatory paragraph could be added to discuss how different growth conditions or environmental factors affect the production and content of heptadec-8-ene.
Answer 3: It is already described in the discussion part that the amount of heptadec-8-ene in brown and green algae is generally very low but was enriched when these algae were damaged mechanically or physiologically [17–19]. In comparison, the percentages of heptadec-8-ene in red algae Pyropia yezoensis, Pyropia haitanensis, and Bangia sp. (Rhodophyta) were 30–50% of the headspace volatile compounds, much higher than in the green alga Ulva australis (formerly Ulva pertusa) (Chlorophyta), the brown alga Sargassum thunbergii and the red alga Gracilariopsis lemaneiformis (formerly Gracilaria lemaneiformis) [20]. From the marine green alga, Bryopsis maxima in a phosphate buffer (Z)-heptadec-8-ene was released by mechanical wounding, but this process was inhibited by heat treatment [17]. Therefore, it was supposed that (Z)-heptadec-8-ene would be also formed enzymatically from a fatty acid and might act as a chemical signal or a semiochemical. There is likely an enzymatic system (a type of heme protein) in Pyropia (Rhodophyta), which catalyzes eicosapentaenoic acid to produce heptadec-8-ene [20] with the optimal conditions of pH 9 and 25 ℃, but the enzyme was thermal resistant with over 50% activity remaining at 60–100 ℃. Although the enzyme inhibitors phenidone, phenanthroline, and L-cysteine had no effect, the enzyme was significantly inhibited by hemoprotein, NaN3, which is consistent with the result obtained for Pyropia tenera and Pyropia sp. (Rhodophyta) [20]. In addition it is possible that different growth conditions and envirromental factors affect the production of this compound, but it is difficult to mention all poosible reasons for the chance of its concentration in the discussion.
Question 4:
- Two fibers of different polarity were used in the paper to collect U. lactuca.what was the basis for the choice of these two specific fibers? Is it possible that other important compounds were missed?
Answer 4: These 2 fibers are usually used for the investigations of the headspace compounds from algae (as in our previous published papers). They differ according to the polarity, but the obtained chemical profiles are complementary. It is not likely that other important headspace compounds were missed and in fact the use of 2 types of fibers enables more reliable results that the use of 1 fiber.
Question 5:
- The text mentions that the seaweed Ulva lactuca was collected in the Adriatic Sea in May by single-point sampling. Does the sampling time cover the key stages in the growth cycle of the seaweed? Does it take into account the effect of seasonal variations on the chemical composition of the seaweed?
Answer 5: Our study was designed as a preliminary investigation to characterize the chemical composition of Ulva lactuca in a specific season, with May being selected due to the high growth rate and metabolic activity of the alga in spring. Due to practical constraints and limited resources, only single-point samples were taken. Although we are aware that seasonal sampling would provide a more comprehensive overview of chemical profile and possible variability, this initial overview provides valuable baseline data and insights that may be useful for future studies with expanded seasonal sampling.
Question 6:
- It is mentioned that U. lactucacontains all essential amino acids except histidine and leucine, which account for almost one-third of the total amino acid content. Does the absence of histidine and leucine limit the potential of U. lactucaas a protein source? Are there ways to supplement these amino acids?
Answer 6: Although the concentrations of Hys and Leu were lower that other EAA, the overall amino acid profile and protein content that was 17.66 % does not reduce the nutritional value of U. lactuca as a protein source. This is supported by literature data that show that other edible macroalgae (e.g. Durvillaea antarctica or Himanthalia elongate) have a lower protein content [35, 38] compared to the U. lactuca. We are also aware that some macroalgae are better source of protein and have higher content of EAA but U. lactuca is one of the most consumed macroalgae and therefore can contribute to total protein and EAA intake along with other nutritionally valuable ingredients. Furthermore, we have stressed in our paper that many external factors (like location, season, water salinity and temperature), and for the amino acid profile additionally extraction procedure and solvent type had a great impact on obtained results. As mention in previous answer, the results presented in this study provide an overview of characteristics and composition of U. lactuca from Adriatic Sea but further research is needed to get better insight into chemical composition of U. lactuca in different seasonal sampling.

Round 2
Reviewer 2 Report
Comments and Suggestions for Authors
please add the correct Table 5
Reviewer 3 Report
Comments and Suggestions for Authors
Accept your modifications.